# Emerging Role of ERBB2 in Targeted Therapy for Metastatic Colorectal Cancer: Signaling Pathways to Therapeutic Strategies

**DOI:** 10.3390/cancers14205160

**Published:** 2022-10-21

**Authors:** Nannan Wang, Yuepeng Cao, Chengshuai Si, Peng Shao, Guoqing Su, Ke Wang, Jun Bao, Liu Yang

**Affiliations:** 1Department of Colorectal Surgery, The Affiliated Cancer Hospital of Nanjing Medical University & Jiangsu Cancer Hospital & Jiangsu Institute of Cancer Research, Nanjing 210009, China; 2Department of Medical Oncology, The Affiliated Cancer Hospital of Nanjing Medical University & Jiangsu Cancer Hospital & Jiangsu Institute of Cancer Research, Nanjing 210009, China

**Keywords:** ERBB2, HER2, targeted therapy, colorectal cancer

## Abstract

**Simple Summary:**

Colorectal cancer (CRC) is the third most commonly diagnosed malignancy and the second most common cause of cancer-related mortality worldwide. Currently available targeted therapies for metastatic CRC mainly target vascular endothelial growth factor and epidermal growth factor receptor in RAS wild-type tumors. Although Erb-B2 receptor tyrosine kinase 2 (ERBB2/human epidermal growth factor receptor 2) plays a significant therapeutic role in breast and gastric cancers, there are no licensed ERBB2-targeted therapies for metastatic CRC. This review aims to outline the molecular biology of ERBB2-positive metastatic CRC and potential targeted therapeutic strategies.

**Abstract:**

Despite recent improvements in the comprehensive therapy of malignancy, metastatic colorectal cancer (mCRC) continues to have a poor prognosis. Notably, 5% of mCRC cases harbor Erb-B2 receptor tyrosine kinase 2 (ERBB2) alterations. ERBB2, commonly referred to as human epidermal growth factor receptor 2, is a member of the human epidermal growth factor receptor family of protein tyrosine kinases. In addition to being a recognized therapeutic target in the treatment of gastric and breast malignancies, it is considered crucial in the management of CRC. In this review, we describe the molecular biology of ERBB2 from the perspective of biomarkers for mCRC-targeted therapy, including receptor structures, signaling pathways, gene alterations, and their detection methods. We also discuss the relationship between ERBB2 aberrations and the underlying mechanisms of resistance to anti-EGFR therapy and immunotherapy tolerance in these patients with a focus on novel targeted therapeutics and ongoing clinical trials. This may aid the development of a new standard of care in patients with ERBB2-positive mCRC.

## 1. Introduction

Colorectal cancer (CRC) is one of the most common causes of tumor-related deaths globally, with 1.85 million cases and 850,000 deaths each year. It ranks third in terms of incidence and second in terms of mortality among all cancer types [1]. Overall, 20% of individuals with CRC have metastatic disease at presentation, and another 25% who initially present with in situ tumors develop metastases during follow-up [2]. The prognosis is poorer for metastatic CRC (mCRC), which has a 5-year survival rate of less than 20% [3]. This is largely because of more aggressive tumor biology and the lack of effective therapeutic strategies.

Except for in a small subset of patients who are suitable for curative surgical excision of metastases, systemic therapy (which is largely palliative) forms the basis of treatment in advanced CRC. Currently approved front-line chemotherapy agents for KRAS/NRAS wild-type (WT) mCRC include traditional cytotoxic remedies (fluorouracil, oxaliplatin, and irinotecan), vascular endothelial growth factor inhibitors (bevacizumab), and epidermal growth factor receptor (EGFR) inhibitors (cetuximab and panitumumab); these biomarker-directed regimens have driven recent improvements in survival [2,4]. Notably, immune checkpoint inhibitors such as pembrolizumab have been approved by the National Comprehensive Cancer Network clinical practice guidelines for the first-line treatment of microsatellite instability-high (MSI-H) or deficient-mismatch repair (dMMR) advanced CRC [5]. In this context, it is important to recognize that even tumors with similar histology demonstrate a considerable degree of molecular heterogeneity, and additional genetic alterations induced by therapeutic pressure ultimately cause disease progression [6].

Despite ongoing efforts to individualize patient care, most drugs used in the treatment of CRC are administered without genetic testing and molecular selection (e.g., cytotoxic and anti-angiogenic drugs). MSI-H/dMMR (usually includes four proteins, namely, MLH1, MSH2, MSH6, and PMS2 for identifying patients suitable for immunotherapy) and RAS gene family mutations (mostly found in exons 2, 3, and 4 of KRAS and NRAS for determining suitability for treatment with anti-EGFR monoclonal antibodies) are the only clinically established predictive markers [7,8]. The activation of Erb-B2 receptor tyrosine kinase 2 (ERBB2) by gene amplification or mutations (occurring in approximately 5% of mCRC) has recently been identified as a cause for resistance to anti-EGFR therapy and tolerance to immune checkpoint inhibitors [4,9]. In addition to being an important therapeutic target in individuals with non-small cell lung, breast, and gastroesophageal cancers [10,11], ERBB2 also demonstrated potential as a therapeutic target for other cancers in several randomized clinical trials; ERBB2 status is therefore considered during planning of combination therapy strategies [12,13,14,15]. Although ERBB2-targeted agents are not routinely administered to patients with mCRC who test positive for the protein, ERBB2 plays a significant role in CRC; it may therefore be a predictor of success for innovative targeted therapies in the age of precision medicine and in the context of genetic heterogeneity [16].

In this review, we provide a comprehensive molecular perspective on the structural features of the ERBB2 receptor, common genetic alterations in ERBB2, and activation of downstream signaling pathways. In particular, we describe the role of ERBB2 in resistance to targeted therapy and tolerance to immunotherapy. In this context, the novel targeted agents and clinical trials on ERBB2 demonstrate its great potential as an emerging actionable target for mCRC.

## 2. Molecular Biology of the ERBB2 Signaling Pathway

### 2.1. Structural Features of the ERBB2 Receptor

The four ERBB (or EGFR) proteins are members of subclass I of the RTK superfamily; they include EGFR/ERBB1/HER1, ERBB2/HER2, ERBB3/HER3, and ERBB4/HER4. These receptors are essential for a variety of cellular processes, including cell growth, proliferation, differentiation, and migration [17]. In addition to an extracellular domain (ECD) or ligand-binding area and a single transmembrane domain (TMD), all receptors comprise a cytoplasmic/intracellular region composed of a juxtamembrane domain (JMD), a kinase domain, and a C-terminal tail domain (Figure 1) [18,19]. The ECD comprises four subdomains (I-IV); in the absence of a ligand, domains II and IV adopt an auto-inhibited tethered (closed) conformation. Upon ligand interaction between domains I and III, the dimerization arm in domain II unhinges, leading to receptor homo- or heterodimerization, allosteric kinase activation, and C-terminal tail domain phosphorylation [19]. This process recruits and activates various downstream signaling proteins containing Src homologous structure-2 (SH2) or phosphotyrosine binding structural domains and engages downstream mediators to drive important cellular signaling pathways [20]. Despite comparable levels of total phosphotyrosine, EGF-activated ERBB2 binds Shc considerably weakly compared to ERBB2 that has been dimerized consequent to mutations in the transmembrane structural domain [21]. Therefore, the signal produced by receptor heterodimers is distinctive; instead of being the simple sum of individual dimer partner signaling proper ties, it reflects the properties of the heterodimers.

Numerous ligands bind to the ECD of the ERBB receptor; these include EGF, neuregulin, and transforming growth factor-α, among others [22,23]. Growth factors that cause receptor dimerization and/or oligomerization upon binding to extracellular areas of the receptor represent receptor-specific ligands which typically activate ERBB family members [24]. However, ERBB2 is a unique member of the ERBB family, as it has a completely different extracellular structural domain from other receptors and no high-affinity ligands. ERBB2 possesses a stable conformation that mimics the ligand-activated state, with an absent domain III-V link and an exposed dimerization loop in domain II [25,26]. Structural domains I and III are in close proximity in the “open” conformation, preventing ligand binding to EGF-related peptides; this structure explains why ERBB2 lacks a ligand [25]. Notably, ERBB2 is unable to bind to any growth factor, making it impossible to induce the production of functional ERBB2/ERBB2 homodimers. Only non-physiological ERBB2 overexpression leads to the production of functional homodimers [27]. All other ERBB family members favor ERBB2 as a dimer partner and have varying signaling capacities [28,29]; in this context, homodimers have lower signaling continuity than heterodimers [30]. ERBB heterodimerization enables the incorporation of kinase-deficient ERBB3 and ligand-free ERBB2 into the signal transduction cascade. Based on this finding, this ERBB pair is believed to function as an oncogenic unit; it is the most effective in terms of contact strength, ligand-induced tyrosine phosphorylation, and downstream signaling [28,31]. Studies indicate that ERBB2 oncogenic activity from tumor ERBB2 overexpression may depend on the presence of ERBB3 [32,33].

Interestingly, ligand-induced ERBB receptor heterodimerization adheres to a rigid hierarchy. The fact that the activation of ERBB3 is hindered in the absence of ERBB2 suggests that the latter plays a part in the lateral transfer of signals between other ERBB receptors [17]. Additionally, a member of the mucin family modifies the location of ERBB2 (particularly of a phosphorylated form) in epithelial cells of the colorectum while acting as an intramembrane regulator of ERBB2 activity; this indicates that it is particularly important in the regulation of ERBB2 signaling [34]. Therefore, although none of the EGF-related peptides directly bind to ERBB2, they all cause heterodimerization and cross-phosphorylation; this in turn leads to tyrosine phosphorylation. The signal flow is accomplished via phosphorylation cascades, which begin with receptor alterations and conclude at the level of specific transcription factors.

### 2.2. ERBB2 Downstream Signaling Pathways

The characteristics of the activating ligand and the heterodimer partner are the most critical elements that decide which of the several downstream adaptor proteins will be engaged, and consequently, which pathway will be activated [30]. The C-terminal tyrosine residues of each ERBB receptor exhibit a distinctive autophosphorylation pattern, which serves as a docking site for SH2 or phosphotyrosine binding domains. Shc, Crk, Grb2, Grb7, and Gab1 are examples of adaptor proteins; Src, Chk, and phosphatidylinositol 3-kinase (PI3K; via the p85 regulatory subunit) are kinases; and SHP1 and SHP2 are protein tyrosine phosphatases. The signaling pathways triggered by the four ERBB receptors have been found to demonstrate considerable overlap. At least two important pathways are engaged in the downstream propagation of active ERBB2 signaling: the RAS/mitogen-activated protein kinase (MAPK)-dependent pathway and the PI3K-dependent pathway (Figure 2) [35]. The activation of ERBB2 signaling initiates multiple coordinated biological reactions, including mitogenesis, apoptosis, cellular motility, angiogenesis, and differentiation regulation [30].

ERBB2 participates in the MAPK signaling pathway, one of the most crucial channels for cell proliferation. The three main subfamilies of MAPK include the extracellular-signal-regulated kinases (ERK MAPK, Ras/Raf1/MEK/ERK), the c-Jun N-terminal or stress-activated protein kinases (JNK or SAPK), and MAPK14 [36]. The MAP kinase enzymes are notable in that dual specificity kinases (also termed MAP/ERK kinases [MEKs] or MAP kinase kinases) need to be activated to phosphorylate both threonine and tyrosine sites [37]. Their activity is regulated by phosphorylation, which enzymatically activates MEKs. The phosphorylation state is tightly controlled by a family of proteins known as the MAP kinases (MAPKKK, MKKK, or MEKK), of which the c-Raf proto-oncogene is the most notable member [37]. Activated MAP kinases phosphorylate and activate transcription factors that are already present in the cytoplasm or nucleus; this results in the expression of certain target genes and a biological response. Responses to moderate outputs are integrated via numerous connections between several MAP kinase cascades.

The PI3K-dependent pathway is another signaling pathway associated with ERBB2. PI3Ks are divided into three classes (I–III) based on their preferred substrates and sequence homology [38]. The various isoforms within each class of PI3K and the various classes of PI3K play specific functions in cellular signal transduction. ERBB2 activates class IA PI3Ks, which are heterodimers containing a p110 catalytic subunit and a p85 regulatory subunit [39]. A lipid second messenger, namely, phosphoinositol-3,4,5-trisphosphate, binds to the pleckstrin-homology domains of numerous downstream molecules to activate them [40]; protein serine/threonine kinase AKT, commonly known as PKB, is one of its primary targets [41,42]. The mammalian targets of rapamycin-rictor kinase complex and 3-phosphoinositide-dependent kinase then recruit phosphoinositol-3,4,5-trisphosphate-bound AKT to the membrane [43], where it is subsequently phosphorylated [44]. These events lead to the full activation of AKT, which in turn phosphorylates multiple target proteins and regulates a number of cellular activities. The forkhead family of transcription factors is a significant AKT target; following phosphorylation by AKT, 14-3-3 proteins sequester them in the cytoplasm, rendering them inactive [38]. The key biological activities of the PI3K-dependent pathway include cell metabolism, cell cycle and cell survival related functions, protein synthesis, cell polarity and motility, and vesicle sorting, among others.

### 2.3. ERBB2 Gene Alterations in mCRC

The ERBB2 gene, which is found on chromosome 17q21, encodes the 185-kDa transmembrane protein ERBB2, also referred to as HER2. Expression of this protein and activation of signaling promotes a number of cellular processes including cell migration, growth, adhesion, and differentiation, that are linked to tumorigenesis [11,18]. Various solid tumors demonstrate different ERBB2 gene alterations including overexpression, amplification, and other mutations. Approximately 7% of patients with CRC have ERBB2 mutations; according to the Cancer Genome Atlas data, these are most frequently found in RAS and BRAF WT tumors [45]. In this context, ERBB2 gene amplification is the typical cause of ERBB2 protein overexpression in 4–5% of mCRC cases [45,46,47]. Although reports are conflicting, ERBB2 overexpression has been observed more commonly in tumors with an advanced T stage and high tumor mutational burden [48]. In this context, some studies have demonstrated a substantial difference between primary tumors and metastases, indicating a decline in ERBB2 positivity with disease progression [49,50]. ERBB2 status is also related to tumor sidedness; the rectum and left colon are frequently the sites of early-stage malignancies with ERBB2 amplification, which most likely results from variables affecting germinal developmental differences [51,52,53]. However, the impact of ERBB2 amplification on the prognosis of mCRC has been debated.

Somatic activating mutations of ERBB2 have also recently been identified as contributors to the development of cancer. These mutations were first noticed in non-small cell lung cancer, followed by a number of other malignancies [54]. In all malignancies, the majority of mutations (46%) occur in the ERBB2 tyrosine kinase domain, which includes exon 20 (20%), exon 19 (11%), and exon 21 (9%) [55]. The ECD harbors 37% of ERBB2 mutations, with those of S310F/Y, Y772dupYVMA, L755P/S, V842I, and V777L/M being the most prevalent [55]. In CRC, ERBB2 mutations are most prevalent in exon 21 (23%) and the ECD (23%); the V842I variation in exon 21 is the most frequent (19%) [55]. In an in vitro study, the introduction of ERBB2 mutations at S310F, L755S, V777L, V842I, and L866M of colonic epithelial cells activated the HER2 signaling pathway and promoted non-anchored cell proliferation; this indicated that these were activating mutations [56].

Diverse ERBB2 mutations have been found in all ERBB2 gene exons including extracellular, transmembrane, or tyrosine kinase cytoplasmic domains. Downstream signaling pathways are therefore activated even in cases with normal gene copy numbers [57]. In their analysis on 111,176 tumors, Pahuja et al. found that the extracellular and kinase domains account for the majority of these changes (approximately 40% each), whereas the TMD and JMD account for 2.8% and 7.7% of the mutations, respectively [58]. In this context, G660D, R678Q, E693K, and Q709L are frequent mutations in the TMD and JMD of HER2 [58]. Pahuja et al. also evaluated the heterozygous germline HER2 TMD mutation (G660D), which was found in an Indian family; those with the mutation experienced early-onset lung cancer [58]. The majority of HER2 somatic mutations result in receptor activation, based on their capacity to boost intracellular signaling, trigger oncogenic transformation, and accelerate the growth of xenograft tumors [59,60]. However, certain HER2 mutants such as V773M have a reduced propensity for cellular activation owing to the lower quantities of phosphorylated HER2 and downstream signaling molecules that result from their synthesis [61].

## 3. Detection of ERBB-Positivity in mCRC

Several tissue/slide-based techniques including immunohistochemistry (IHC), fluorescence in situ hybridization, polymerase chain reaction, and next generation sequencing (NGS), can be used to evaluate ERBB2 overexpression and/or amplification [62]. IHC is an antibody-independent, semi-quantitative approach that is used to identify the membrane-bound HER2 receptor in formalin-fixed paraffin-embedded tissue samples. The diagnostic criteria of the HERACLES study defined HER2 positivity as 3+ (expression in less than 50% of cells), 2+ (moderate positivity with expression in less than 50% of cells), or 3+ expression in more than 10% but less than 50% of tumor cells. Staining intensities of 0+ and 1+ were regarded as negative [63]. It is worth noting that approximately one-third of mCRCs with ERBB2 alterations only contain somatic variations, which are undetectable by standard IHC and fluorescence in situ hybridization testing. In addition, HER2 expression is less heterogeneous in CRC than in gastric cancer, where the suitability of the interpretation criteria is not well defined. More sensitive and precise approaches are therefore required for identifying ERBB2 alterations in CRC. In addition to evaluation of tumor tissue biopsy specimens, ERBB2 amplification can be identified by non-invasive methods [9], including assessment of ERBB2 overexpression in circulating tumor cells or measurement of ERBB2 copy numbers via NGS of cell-free DNA. In another study, comprehensive circulating tumor deoxyribonucleic acid (ctDNA) NGS successfully detected ERBB2 amplification in 96.6% (28/29) of the intent-to-treat group; this indicates that ctDNA can serve as a substitute for tissue [64]. Additional methods have been considered for the identification of elevated ERBB2 activity; these include measurement of ERBB2 messenger ribonucleic acid expression levels [65] and thorough genomic changes analysis of DNA recovered from formalin fixed paraffin embedded samples [66,67,68,69,70], among others. However, when paired with the data published in the Cancer Genome Analysis and Catalogue of Somatic Mutations in Cancer databases, these studies showed a nearly equivalent frequency of ERBB2 amplification and short variant mutation [71].

The diagnosis of ERBB2 positivity in CRC remains controversial. ERBB2 heterogeneity is one of the factors affecting diagnosis; it is characterized by differential expression or amplification within the same tumor (intratumoral) or in cancers from various sites or time points in the same patient (intertumoral) [72]. The phenomenon has been described in literature on breast and lung cancer [73,74,75,76]. A retrospective study showed a significant degree of discordance between matched pairs of primary tumors and metastases, with an HER2-positivity rate of 11.2%, 10.1%, and 31.8% in primary CRC tumors, matched positive lymph nodes, and corresponding metastases, respectively [50]. Heterogeneous expression or amplification in malignancies pose a challenge in ERBB2 diagnostics, as the selection of patients for anti-ERBB2 therapy depends on the overexpression of the ERBB2 protein or amplification of the ERBB2 gene [77]. The dynamic variations of ERBB2 during tumor progression and treatment are also unclear; this includes the manner in which chemotherapy could influence ERBB2 expression [78]. A study on advanced malignancies found that HER2 amplification and overexpression recurs in distant metastases; this suggests the occurrence of genetic differentiation as carcinoma in situ progresses to invasive carcinoma [79]. Ongoing ERBB2 expression monitoring is therefore needed to assess the feasibility of continuation of ERBB2-targeted drugs following changes in treatments or disease characteristics. With the current findings, non-invasive assays such as liquid biopsy may gradually become the mainstream method for detecting HER2 positivity in the future, as it allows dynamic monitoring of tumors as they undergo drug pressure or disease progression. However, these methods are still in the research phase. Therefore, the results of more prospective clinical studies with large samples are expected to provide pre-treatment evidence for ERBB2 detection along with RAS, BRAF, and MSI-H.

## 4. ERBB2-Targeted Treatments in mCRC

### 4.1. Impact of ERBB2 Activation on Anti-EGFR Treatment Resistance

Resistance to EGFR monoclonal antibodies has considerably compromised the clinical use of targeted agents such as cetuximab and panitumumab; it also hinders the improvement of prognosis in ERBB2-positive mCRC [80,81]. A proportion of patients with mCRC and RAS WT tumors experience resistance to anti-EGFR therapy as a result of ERBB2 signaling, which activates the EGFR compensatory feedback loop [56,82,83]. Studies also demonstrate that ERBB2-amplified CRC may benefit from combined treatment with MEK and PI3KCA inhibitors [84]. The introduction of major ERBB2 mutations (including S310F, L755S, V777L, V842I, and L866M) in CRC cell lines has been found to maintain MAPK phosphorylation, which leads to resistance to cetuximab and panitumumab [56]. A large preclinical study demonstrated genotype-therapeutic response correlation with ERBB2 amplification in cetuximab-resistant KRAS/NRAS/BRAF/PIK3CA WT subgroups; it also found that patients with ERBB2 amplification exhibit significant and durable tumor regression after combined inhibition of ERBB2 and EGFR [82]. Preliminary clinical data have also revealed that patients with mCRC carrying amplified or overexpressed ERBB2 molecular features have poorer objective tumor response, overall survival, and progression-free survival following EGFR targeted therapy [85,86]. This shows that in CRC, the initial resistance to EGFR may be driven by activation of the ERBB2 signaling pathway.

Some studies have partially revealed potential mechanisms of resistance to anti-EGFR therapy through transcriptional analysis of ERBB2 activation or patient-derived xenograft cells (PDXs). A recent study found that increased ERBB2 expression in cd44v6-positive CRC stem cells is associated with activation of the PI3K/AKT pathway, which encourages acetylation of the ERBB2 gene regulatory area in these cells [87]. Cytokines released from tumor-associated fibroblasts promote resistance of CRC stem cells to PI3K/AKT inhibitors. Triple targeting of ERBB2 bound to PI3K and MEK inhibitors leads to CRC stem cell death and degeneration in transplantation tumors (Figure 2); this includes those carrying KRAS and PIK3CA mutations [87]. A recent study offered compelling evidence in favor of a pan-HER receptor inhibition strategy in KRAS-WT CRC for overcoming resistance to anti-EGFR therapy. In this study, a triple HER receptor inhibitor (including a newly developed antibody that targets both EGFR and HER3 receptors and the classical HER2 inhibitor trastuzumab) successfully inhibited heregulin-induced HER receptor phosphorylation and downstream signaling in CRC cell lines. Notably, the growth inhibition demonstrated by triple HER targeting was replicated in primary KRAS WT-derived organ cultures exposed to heregulin [88]. In their study, Lupo and colleagues found that mCRC cells that survived EGFR inhibition displayed gene expression patterns comparable to those of Paneth cells in a quiescent fraction of healthy intestine secretory precursors [89]. These pseudodifferentiated tumor remnants demonstrated decreased expression of EGFR-activating ligand-encoding genes, increased HER2 and HER3 activity, and persistent PI3K pathway activation. After EGFR neutralization, residual tumor reprogramming is mediated mechanistically by inactivation of yes-associated protein, an important regulator of intestinal epithelial cell damage recovery. In this context, pan-ERBB antibodies decreased residual lesions, inhibited PI3K signaling, and promoted long-term tumor control in preclinical trials [89]. Tolerance to EGFR inhibition is characterized by deactivation of an intrinsic lineage program that promotes regenerative signaling and EGFR-dependent carcinogenesis during intestinal repair. In conjunction, these findings contribute significantly to the understanding on intrinsic and extrinsic mechanisms linking ERBB2 overexpression to resistance to anti-EGFR drug therapy. In addition, these findings offer new therapeutic perspectives regarding the management of resistance to EGFR-targeted agents and may provide new opportunities for ERBB2-based clinical trials.

The molecular mechanisms of primary resistance to anti-EGFR drugs may play an important role in acquired resistance. A study found that ERK 1/2 signaling persisted after activation of ERBB2 signaling owing to ERBB2 amplification or neuromodulin overexpression in cell lines; this resulted in cetuximab resistance. In addition, both in vitro and in vivo suppression of ERBB2 or disruption of ERBB2/ERBB3, heterodimerization restored cetuximab sensitivity [90]. This provides important evidence to demonstrate the role of aberrant ERBB2 signaling in primary and acquired drug resistance among patients with CRC who receive anti-EGFR therapy (cetuximab). In this context, ERBB2 amplification has indeed been found to be associated with selective pressure of anti-EGFR therapy [9]. The findings from a study suggested that although HER2 amplification may be absent in tumors not treated with anti-EGFR drugs, it is elevated after exposure to these agents [91]. There has been considerable debate regarding certain ideas that lack solid clinical evidence. Some studies have shown that ERBB2 expression is significantly elevated following the use of EGFR drugs. The underlying cause of this phenomenon is unclear; it may result from drug pressure, which generates a dominant subclone of amplified or mutated ERBB2 and eventually leads to the development of a drug-resistant phenotype. Therefore, it may be necessary to retest the ERBB2 status after the emergence of acquired resistance to EGFR; this may allow estimation of the potential benefit from anti-EGFR, anti-ERBB, or multi-targeted therapy. A biomarker-based liquid biopsy (e.g., ctDNA) may play a more important role than tissue biopsy. It may indicate the benefits of rechallenge with anti-ERBB or EGFR targeted therapy and combination therapy; it may also aid the design of personalized regimens for patients with RAS-WT tumors.

### 4.2. Role of ERBB2 Positivity in mCRC with Immunotherapy Tolerance

Immunotherapy has become an important treatment option in CRC. The findings from the KEYNOTE177 study have prompted the use of pembrolizumab in the first-line for MSI-H or dMMR mCRC; the Chinese Society of Clinical Oncology guidelines for CRC have also been revised accordingly [92]. However, only 5% of patients with advanced mCRC have MSI-H tumors; the objective remission rate with programmed death protein 1 (PD-1)/programmed death-ligand 1 antibodies alone or in combination with other drugs is only approximately 40% [93]. There has been considerable discussion regarding immunotherapy tolerance in mCRC. Evidence suggests that aberrant signaling by ERBB family members plays an important role in the development of multiple malignancies and anti-tumor immune escape [94]. Extensive immunogenomic analysis of data from the Cancer Genome Atlas has revealed that HER amplification appears to be associated with a non-inflammatory tumor microenvironment [95]. During evaluation of the therapeutic benefits of anti-ERBB2 therapy, several lines of evidence indicate the role of immunological mechanisms including antibody-dependent cell-mediated cytotoxicity to be more significant than that of intracellular signaling. As recently indicated by the positive results from the KEYNOTE-811 trial, harnessing the immune effects of anti-ERBB2 therapies can improve understanding on these mechanisms and elucidate the efficacy of combination immunotherapy and anti-ERBB2 therapies [96]. HER2 amplification results in decreased phosphorylation of TANK-binding kinase 1 and suppresses stimulation of interferon genes signaling; this impairs the interferon and antitumor immune responses (Figure 2) [97]. Overexpression of HER2 in mCRC could also be a potential target for chimeric antigen receptor T cell (CAR-T) therapy. In a study using a NOD scid gamma mouse tumor-bearing model, HER2-specific CAR-T cells exhibited strong cytotoxic and cytokine secretory capacity against CRC cells in vitro and effectively prevented the progression of CRC; notably, this phenomenon was even stronger in PDX models. The findings from this study suggest that HER2 CAR-T cells represent a novel therapeutic approach for mCRC [98]. As normal tissues expressing low levels of HER2 target antigens cross-react with tumor tissues in encounters with CAR-T cells, researchers designed a circuit in which a low-affinity synthetic HER2 Notch receptor controlled the expression of a high-affinity HER2 CAR. T cells with this circuit were able to clearly distinguish normal cells from HER2 overexpressing tumor cells in vivo and in vitro [99].

In this context, tumor vaccines are needed as an additional immunotherapy option; carcinoembryonic antigen and Her2/neu peptide vaccination has been found to demonstrate immunogenicity in patients with advanced CRC and is well tolerated [100]. The clinical efficacy of PD-1 blockade has also been demonstrated in HER2-amplified tumors; however, despite the large number of clinical trials and new agents in development, its impact on HER2 amplified breast and gastric cancers is limited [101,102]. Such studies are lacking in CRC; trials testing the clinical efficacy of immunotherapy in combination with HER2-targeted therapy will therefore be needed. The main challenge of HER2-targeted monoclonal antibody therapy lies in the optimal exploitation of levels and locations of tumor HER2 protein overexpression [103]. Unlike in breast cancer, where HER2 is usually expressed on the cell surface [104], HER2 overexpression rates in CRC are unclear; this may be attributed to different technical approaches, antibodies, and scoring criteria [105]. Immunohistochemical evaluation alone is limited by the inability to distinguish the location of the protein in the cell. On categorization based on membrane and cytoplasmic HER2 overexpression, a distinct pattern is observed in CRC; approximately 5% of patients demonstrate membranous overexpression whereas approximately 30% have cytoplasmic overexpression [106]. This may be an attractive option that warrants further study.

### 4.3. Novel Drugs and Clinical Trials for ERBB2-Positive mCRC

In recent years, several innovative agents have been used to treat ERBB2-positive malignancies, including antibody-drug conjugates, dual-targeted antibodies, tyrosine kinase inhibitors, and ERBB2-targeted immunotherapy [107]. Figure 1 illustrates the mechanism of action of these ERBB-targeting agents. In addition, Table 1 summarizes the pivotal and ongoing clinical trials on ERBB2-positive mCRC and their results based on drug type.

Based on the idea of antibody delivery of cytotoxic drugs to antigen-expressing tumors, Yusuke and colleagues developed trastuzumab deruxtecan (also known as DS-8201a), an HER2-targeted antibody-drug conjugate, in 2016. It is structurally composed of a humanized anti-HER2 antibody, an enzymatically cleavable peptide linker, and a novel topoisomerase I inhibitor [108]. In the PDX model, DS-8201a showed strong antitumor activity against cancers that were initially resistant to trastuzumab emtansine, an antibody-conjugated microtubule protein depolymerizer (trastuzumab-maytansinoid); this may be attributed to the effectiveness of trastuzumab-deruxtecan against tumors with low HER2 expression [108]. DS-8201a exposure was found to strengthen anti-tumor immunity in xenograft mice models and clinical trials. In particular, trastuzumab deruxtecan was able to increase the expression of CD86 on bone marrow-derived dendritic cells in vitro, boost the quantity of tumor-infiltrating CD8+ T cells, and improve the expression of PD-L1 and major histocompatibility class I in tumor cells [109]. The safety and therapeutic efficacy of trastuzumab deruxtecan was assessed in a phase I clinical study (NCT02564900) that included 60 patients with pretreated HER2-expressing (IHC ≥1+) solid tumors other than breast and gastric malignancies. In the six tumor types including CRC, the objective response rate and median progression-free survival were found to be 28.3% (17/60) and 7.2 months, respectively [110]. The phase II DESTINY-CRC01 clinical trial evaluated the antitumor activity and safety of trastuzumab deruxtecan in patients with HER2-expressing mCRC, who had previously progressed on two or more prior treatment regimens and had RAS and BRAFV600E WT tumors; objective remission was observed in 24 (45.3%) of 53 (68%) patients after a median follow-up of 27.1 weeks. At least 10% of individuals experienced grade 3 or higher serious treatment-related adverse effects; interstitial pneumonia or lung-related illnesses were the only cause of mortality thought to be associated with the treatment [15]. However, a comprehensive review that included 14 trials and a total of 1193 patients with various advanced solid malignancies found that interstitial lung disease and pneumonia are significant and possibly fatal adverse events associated with trastuzumab deruxtecan [111]. Although HER2 expressing/mutated solid tumors may benefit from the drug, more research is needed to determine the risk factors and underlying pathophysiological mechanisms for these adverse effects. This will facilitate the design of effective management strategies to halt the progression of these conditions and their symptoms.

The benefit of ERBB2-targeted therapy is now only available to patients with mCRC that is RAS wild-type, according to numerous studies [12,69]. In line with preclinical evidence showing resistance to dual-targeted HER2 in the presence of KRAS mutation, [56] MyPathway showed that trastuzumab + patuximab failed to respond therapeutically in KRAS-mutated, HER2-amplified CRC [13]. One case report, however, describes a 58-year-old man with HER2-amplified mCRC and a KRAS G12D mutation whose disease progressed despite receiving all conventional cytotoxic therapies as well as dual HER2 targeting using trastuzumab and pertuzumab. Subsequently, he achieved clinical benefit in the regression of metastatic lung disease with trastuzumab emtansine (T-DM1) [112]. This case demonstrates the therapeutic potential of anti-HER2 ADCs for RAS-mutant tumors. Although this patient eventually developed liver metastases, this patient achieved a best response of 4 months PFS compared to the median PFS of only 1.4 months reported in the MyPathway trial.

Compared to larger molecules such as antibodies and their derivatives, smaller peptide molecules appear more attractive due to their lower production cost, simplicity, chemical stability, and flexibility in the choice of carrier surface coupling procedures [113]. One of these peptide sequences, KCCYSL (called P6.1), was found to specifically recognize and bind the extracellular domain of HER2, and its monomeric, dimeric, and tetrameric forms can be used to develop targeted delivery systems towards tumor cells overexpressing the HER2 receptor. It has been demonstrated that the uptake of P6.1 peptide tetramer-modified liposomes in tumor cells is similar to that of HER2 antibodies such as Herceptin. Another targeting strategy has been devised by linking metal-organic complexes to stearate-modified polymeric chitosan nanoparticles via KCCYSL to induce ROS, DNA damage, and mitochondrial membrane depolarization and rupture, finally leading to apoptosis of receptor-expressing tumor cells. In vivo experiments showed that this targeted nanomedicine could improve survival rate, reduce tumor volume, and has a good safety profile in mice [114].

Tumor-targeted nanoparticle-based drug delivery systems are one of the important research areas for novel antitumor drugs. Nanoparticles can be used as nanovehicles for specific targeting of tumor cells or tissues to overcome the issues of rapid plasma clearance, poor targeting ability, and systemic toxicity. Specific tumor-targeted drug delivery of antibody-coupled nanoparticles can be achieved by modifying suitable antibodies or their fragments, which are characterized by sophisticated selectivity, high recognition efficiency, and large diversity of antibodies [115]. In addition, the drug-to-antibody ratio of antibody-coupled nanoparticles can be easily adjusted and has a higher concentration of drug internalization than ADCs [115]. A study developed biodegradable antibody-coupled polymeric nanoparticles (NPs) using tamoxifen (Tam), poly (D, lactic-co-glycolic acid) (PLGA), and poly (vinyl alcohol-pyrrolidone) (PVP) coupled with Herceptin antibodies for targeted drug delivery and sustained release against breast cancer cells. This nanovector drug delivery system (DDS) not only actively targets multidrug resistance (MDR)-associated phenotypes (HER2 receptor overexpression), but also improves therapeutic efficiency by enhancing cancer cell-targeted delivery and sustained release of therapeutic agents [116]. Antibody-conjugated nanoparticles have already emerged in the pharmaceutical industry, contributing to safer and more effective applications in drug delivery, imaging, and immunotherapy.

Zanidatamab (ZW25) is a bispecific antibody that binds to two HER2 epitopes simultaneously developed by Zymeworks; ECD4 is the binding domain of trastuzumab and ECD2 is the binding domain of pertuzumab. Preclinical studies have shown that ZW25 stimulates the immune system and inhibits HER2 signaling more effectively than trastuzumab or pertuzumab in the range of HER2 expression for greater anti-tumor activity [117]. Currently, 24 HER2-positive non-breast cancer patients, including five patients with CRC, are participating in a phase I basket trial to test the medication. The median progression-free survival for these patients, who had received an average of three prior treatments, was 6.2 months, and there were no side effects of grade 4 or 5. This trial demonstrates the security and effectiveness of bispecific antibodies for the treatment of solid tumors that are HER2-positive [117]. Zymeworks has also developed an antibody-drug conjugate, ZW49, which is based on Zanidatamab in combination with a novel microtubule inhibitor, N-acyl sulfonamide auristatin [118]. In a phase I clinical trial, ZW49 is being evaluated for its safety and tolerability in patients with locally advanced or metastatic HER2-expressing cancers (NCT03821233). Another bispecific antibody, A166, binds a cytotoxic anti-microtubule drug called duostatin-5 (Duo-5) to an antibody that shares the same amino acid sequence as trastuzumab. In a phase I clinical trial, its safety profile was noted, and two patients in the cohort with HER2-positive CRC had an objective response rate of 100%. The drug is currently under research for approval in HER2-positive/amplified relapsed or refractory tumors, including CRC (NCT03602079).

Numerous emerging TKIs such as neratinib, tucatinib, sapitinib, pyrotinib, and poziotinib have stimulated great interest in the development of different novel HER2-targeted therapeutic strategies. Neratinib is an irreversible, orally bioavailable small molecule inhibitor that blocks the activation of downstream signal transduction pathways by binding to and inhibiting EGFR, HER2, and HER4 [119]. It is now a standard of care for patients with HER2-positive early-stage breast cancer as an adjuvant treatment [120]. NSABP FC-7, a recent phase Ib study (NCT01960023) combining Neratinib with Cetuximab for Cetuximab or Panitumumab-resistant Quadruple-WT (KRAS, NRAS, BRAF, PIK3CA)-type mCRC, showed good tolerance of neratinib but no objective responses were observed: seven patients obtained stable disease, four of whom had ERBB2 amplification either in their primary tumor or in the enrollment biopsy [121]. In a phase II clinical trial, 11 patients with mCRC and ERBB2-positivity were given trastuzumab and pyrotinib treatment [122]. In this study, 50% of patients with mCRC were KRAS wild-type, and the objective remission rate was 27%. The most frequent grade III adverse event was diarrhea (73%), which led to dosage stoppage and reduction in 64% of patients, respectively [122]. Tucatinib, an oral, small-molecule, selective HER2 inhibitor, binds reversibly to the ATP pocket of the HER2 receptor’s internal domain and inhibits the activation of ERBB2 signaling pathways and the connections between HER2 and HER3 that result in apoptosis evasion [123]. In 2020, tucatinib gained its initial approval in the USA for treatment in patients with HER2-positive advanced, incurable, or brain metastatic breast cancer who have received one or more prior anti-HER2-based therapies [124]. In the MOUNTAINEER trial (NCT03043313), an open-label, phase II trial, 26 patients with HER2-amplified mCRC who were treated with tucatinib and trastuzumab achieved an ORR of 52% (12/23 evaluable patients), a median PFS of 8.1 months, and an OS of 18.7 months. However, after a median follow-up period of 10.6 months, the median duration of response had not yet been reached [125]. Given the frequency of HER-2 overexpression in CRC and the evidence of clinical success with tucatinib in treating breast cancer, continued study of tucatinib as a monotherapy and in combination with other treatment agents is still of interest.

With the development of immunotherapy for tumors, a number of ongoing clinical studies are now opening up new ideas for the targeted treatment of HER2-positive CRC and investigating the safety and efficacy of HER2-targeted vaccines, chimeric antigen receptor (CAR) treatments, and allogeneic donor cell immunotherapies. Two chimeric ERBB2 vaccines (pertuzumab-like and trastuzumab-like) were tested in patients with a range of metastatic solid tumors, including mCRC, to determine their effectiveness and safety (NCT01376505). Multiple ERBB2-positive solid tumors, including mCRC, were subjected to an anti-ERBB2 chimeric antigen receptor (CAR)-modified T-cell therapy evaluation (NCT02713984) [126]. Another phase I trial used trastuzumab or cetuximab in conjunction with allogeneic donor-derived natural killer (NK) cell cancer immunotherapy (FATE-NK100) to treat multiple ERBB2-positive tumors. (NCT03319459). In a preclinical investigation, the anti-HER2 CAR-macrophage CT-0508 produced an inflammatory microenvironment. The effectiveness and safety of it in treating solid tumors that overexpress HER2, such as CRC, are now being investigated (NCT04660929) [127]. Additionally, intra-tumor injections of HER2-AdVST were combined with CAdVEC, an adenovirus that encourages immunization coupled with ERBB2 chimeric antigen receptor-modified adenovirus-specific cytotoxic T lymphocytes. An oncolytic virus that strengthens the immune system called CAdVEC is now being studied in a clinical trial (NCT03740256).

In conclusion, although HER2 overexpression in CRC is only found at a low rate, encouraging results have been demonstrated in several pivotal clinical studies targeting HER2-positive CRC patients with HERACLES, MyPathway, and DESTINY-CRC01, etc. In addition, a number of novel targeted or immunologic agents and their combination therapies have shown initial efficacy in phase I clinical studies. In the future, as studies continue, phase III clinical studies may provide sufficient evidence for the use of anti-HER2 therapies in this subset of patients and ultimately improve long-term survival in this patient population.

## 5. Perspectives

The management of mCRC is becoming increasingly challenging in the era of precision medicine. Although anti-ERBB2 therapy is not currently recommended as standard treatment for mCRC, it has shown good efficiency and tolerability in preclinical and early clinical studies. As several tumor subtypes are oncogene-dependent, it is necessary to find molecular indicators that will facilitate the use of anti-tumor agents in patients who may obtain maximum benefit. As ERBB2 is one of the most promising oncological targets, methods for its detection need to be enhanced in CRC (along with the criteria for its positivity). HER2-targeted medications frequently result in the gain or loss of HER2 proteins following disease progression. Therefore, liquid or repeat tissue biopsies need to be performed during disease progression and the course of treatment to reassess HER2 status and predict potential therapeutic benefits.

Drug resistance represents a significant challenge in precision medicine. ERBB2 mutations have been demonstrated to bypass suppression of EGFR-dependent signaling. Preliminary findings from preclinical studies that used combinations of prospective medications (such as PI3K and MEK inhibitors) targeting several (or important) downstream cascade signaling pathway components after ERBB2 activation have shown promise; additional validation via clinical trials will be necessary.

Innovative medications that target ERBB2 represent a promising development. Preclinical research and phase I clinical studies have demonstrated the safety of a number of medications based on monoclonal antibodies and small molecule inhibitors. Further phase II clinical studies with larger sample sizes are needed to ascertain their safety and effectiveness in the intended patient population. The presence of HER2 overexpression in cancer cells provides an additional option for the use of CAR-T and cancer vaccines. The best approach for adding these novel agents to the current standard of care (including surgery and chemotherapy), and the optimal order and timing of administration, need to be evaluated in future research.

## 6. Conclusions

Targeted therapeutic strategies for mCRC currently represent an area of active research. Despite limited incidence regarding adverse effects, a substantial number of clinical trials indicate that ERBB2 has the potential to become a new therapeutic target and biomarker in mCRC. Trastuzumab has now been approved for use with chemotherapy as a first-line treatment option for advanced gastric cancer. As numerous novel drugs continue to be developed, it is expected that ERBB2-targeted therapy may be used for mCRC in the near future.

## Figures and Tables

**Figure 1 cancers-14-05160-f001:**
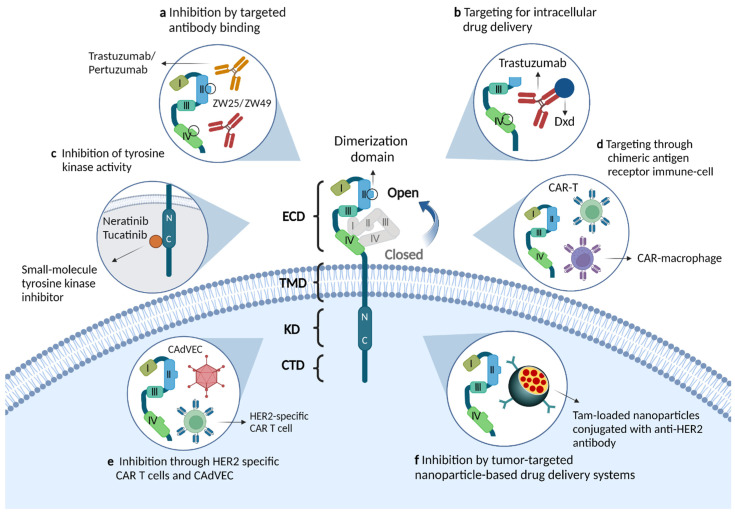
Molecular structure of the ERBB2/HER2 receptor and the mechanism of action of novel therapeutic agents. (**a**) Monoclonal antibodies that target binding to the extracellular structural domain of HER2, including single-epitope monoclonal antibodies (trastuzumab, pertuzumab) and bispecific HER2 antibodies (ZW25, ZW49). The antitumor effects of these drugs are mediated through a variety of mechanisms: inhibition of downstream signaling pathways, involvement in antibody-dependent cellular cytotoxicity or inhibition of receptor dimerization. (**b**) Antibody-drug conjugates targeting HER2, such as trastuzumab emtansine and trastuzumab deruxtecan, release and deliver cytotoxic drugs while binding antibodies and thus exert anti-tumor effects. (**c**) Small molecule inhibitors, such as neratinib and tucatinib, inhibit activation of the PI3K pathway by binding to the tyrosine-kinase domain of the HER2 receptor. (**d**,**e**) HER2-targeted immunotherapies, such as HER2-specific chimeric antigen receptor immune cells or modified oncolytic virus, enhance immune cell activity and stimulate anti-tumor immune responses. (**f**) Tumor-targeting nanoparticles, such as nanodelivery systems with internal encapsulation of tamoxifen and external loading of anti-HER2 antibodies, can efficiently kill HER2 overexpressing breast cancer cells by penetrating the tumor microenvironment.

**Figure 2 cancers-14-05160-f002:**
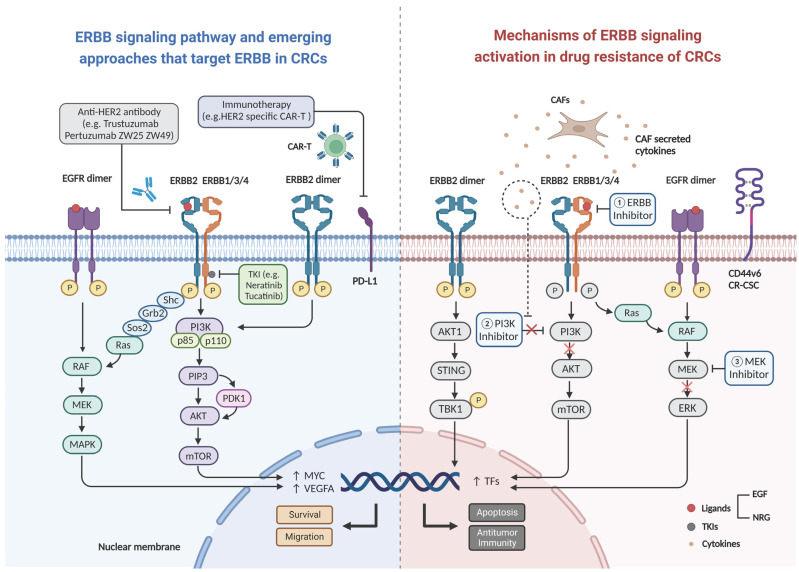
Schematic diagram of ligand binding activation of the ERBB2 signaling pathway leading to drug resistance. Uncontrolled formation of homo- or heterodimers, upon ERBB2 activation under physiological conditions, leads to the activation of two key downstream signaling pathways (the MAPK and PI3K-Akt pathways) and initiation of a series of cellular processes such as survival and migration. Mechanism of inhibition of the signaling pathway by various novel ERBB2 drugs (**left**), and the mechanisms of resistance to anti-EGFR therapy and immunotherapy due to activating of ERBB signaling pathway (**right**). Abbreviations: CAR-T: chimeric antigen receptor T-cell, CAFs: cancer associated fibroblasts, PD-L1: programmed cell death ligand 1, EGF: epidermal growth factor, NRG: neuregulin, CR-CSC: colorectal cancer stem cell, VEGFA: vascular endothelial growth factor A, TFs: tumor factors.

**Table 1 cancers-14-05160-t001:** Clinical trials targeting ERBB2-positive mCRC.

Types of Drugs	Trial/Phase	Interventions	Description	Outcomes/Status
Anti-HER2 mAbs	My Pathway (phase II)	Trastuzumab plus pertuzumab	KRAS-unselected, chemorefractory, HER2 amplified mCRC (*n* = 56); HER2 positivity assigned based on IHC (3+ staining), FISH (ERBB2:CEP17 > 2.0) and/or NGS (ERBB2 copy number gain)	mOS 11.5 months, mPFS 2.9 months, ORR 32%
	TRIUMPH (phase II)	Pertuzumab plus trastuzumab	HER2 amplification mCRC (*n* = 30); ERBB2 amplifications determined using tissue and/or ctDNA analysis	mOS 10.1 months (tissue+) and 8.8 months (ctDNA+), mPFS 4.0 months (tissue+) and 3.1 months (ctDNA+), ORR 30% (tissue+) and 28% (ctDNA+)
Anti-HER2 mAbs+TKIs	HERACLES-A (phase II)	Trastuzumab plus lapatinib	KRAS-WT, chemorefractory, HER2-positive mCRC (*n* = 27); HER2 positivity (HERACLES pathological criteria)	mOS 10 months, mPFS 4.7 months, ORR 28%
Anti-HER2 mAbs+TKIs	MOUNTAINEER (phase II)	Tucatinib plus trastuzumab	RAS-WT, chemorefractory, HER2 amplification/overexpression mCRC (*n* = 26); HER2 positivity determined using IHC (3+ or 2+ staining and FISH-positive), FISH and/or NGS	mOS 17.3 months, mPFS 6.2 months, ORR 55%, DCR NR
	NCT04579380 (phase II)	Tucatinib plus trastuzumab	HER2 overexpression/alterations, unresectable, or metastatic solid timors (*n* = 270, including CRC); HER2 status detected in fresh or archival tumor tissue or blood	Recruiting
	NCT04380012 (phase II)	Pyrotinib plus trastuzumab	HER2-positive advanced CRC(*n* = 40); HER2 positivity confirmed by IHC (3+ or 2+ in more than 50% of cells) and SISH/FISH (HER2:CEP17 > 2.0)	Recruiting
Anti-HER2 ADCs	HERACLES-B (phase II)	T-DM1 plus pertuzumab	RAS-/BRAF-WT, chemorefractory and HER2+mCRC (*n* = 31); HER2 positivity (HERACLES pathological criteria)	ORR 80%, mOS 4.1 months, DCR 9.7%
	DESTINY-CRC01 (phase II)	T-DXd (DS-8201a)	RAS-/BRAF V600E-WT, disease progression on two or more prior regimens, HER2 expressing mCRC (*n* = 78); HER2 positivityCohort A: HER2 IHC 3+ or IHC 2+ staining and ISH-positive (*n* = 53) Cohort B: IHC 2+ staining and ISH-negative (*n* = 7) Cohort C: IHC 1+ staining (*n* = 18)	ORR 83%, mPFS NR, mOS NR, DCR 45.3%
TKIs+anti-EGFR therapy	NSABP FC-7 (phase Ib)	Neratinib plus cetuximab	Resistant to Cetuximab or Panitumumab and quadruple-WT (KRAS, NRAS, BRAF, PI3KCA) disease (*n* = 21); HER2 amplification assessed by CISH (ERBB2:CEP17 > 2.0) or NGS (ERBB2 copy number > 2.0)	RP2D of neratinib: 240 mg/day; no OS; SD was seen at all neratinib doses
Dual-targeted antibodies	NCT03929666 (phase II)	ZW25 plus standard QT	Unresectable, locally advanced, recurrent, or metastatic HER2-expressing cancers (*n* = 362, including CRC); HER2 expressing (IHC 3+ with or without gene amplification; or IHC 0, 1+ or 2+ with gene amplification)	Recruiting
	NCT03821233 (phase I)	ZW49	Locally advanced (unresectable) or metastatic HER2-expressing cancers (*n* = 174)	Recruiting
	NCT03602079 (phase I/II)	A166	Relapsed/refractory, HER2 expressing/amplified cancers (*n* = 49, including CRC); Low HER2 expression (IHC 1+ and IHC 2+ without FISH confirmation) and HER2 positive (IHC2+ with FISH confirmation and IHC 3+)	Active, not recruiting
Anti-HER2 CAR- macrophages	NCT04660929 (phase I)	CT-0508	HER2 overexpressing solid tumors (*n* = 18, including CRC)	Recruiting
CAR-T+Oncolytic adenovirus	NCT03740256 (phase I)	CAdVEC	Advanced HER2 positive solid tumors (*n* = 45, including CRC); HER2 positivity defined as IHC 3+ or IHC 2+ staining	Recruiting

Abbreviations: mAbs: monoclonal antibodies, ADCs: antibody-drug conjugates, TKIs: tyrosine kinase inhibitors, ctDNA: circulating tumor DNA, m: month, mPFS: median progression-free survival, ORR: overall response rates, PR2D: recommended phase 2 dose, SD: stable disease, HR: hazard ratio, NR: not reported, QT: chemotherapy, T-DM1: trastuzumab emtansine, TD: trastuzumab deruxtecan, CAR: chimeric antigen receptor, WT: wild type.

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
