# Peer review of "Emerging Role of ERBB2 in Targeted Therapy for Metastatic Colorectal Cancer: Signaling Pathways to Therapeutic Strategies"

_cancers, 2022, doi:10.3390/cancers14205160_

Round 1

Reviewer 1 Report

The authors provided a very comprehensive overview about the functional ERBB2 pathway, downstream signaling and current options of targeted therapies - in my opinion, the authors should be commended for their efforts.

Nevertheless, in my feeling, some issues should be additionally addressed in this manuscript:

- What would - in the opinion of the authors - be the appropriate approach of routine testing for HER2/neu positivity? New data from DESTINY and also HERACLES and MyPathway suggested high response rate and long survival for these patients, meaning that second-line treatment with anti-HER2/neu drugs should be aimed for. Should we perform immunohistochemistry/FISH before treatment start together with RAS, BRAF, MSI-H, should we test as soon as we know RAS wild-type status, or even after progression on anti-EGFR treatment?

- A statement should be included for detection of HER2/neu positivity by immunohistochemistry that there are diagnostic recommendations derived from study protocols (e.g. HERACLES), but there is not a defined HER2/neu score for mCRC yet.

- It should also be mentioned that HER2/neu inhibition in RAS mutated tumors is significantly less effective compared to RAS wild-type tumors.

Reviewer 2 Report

The authors have written a very well summarised ERBB2 targeted therapy for metastatic colorectal cancer including the drugs and clinical trials of antibody and immunotherapy based medicines. The figure 1 here presents a very well depicted molecular structure and mechanism of the therapeutic agents such as chimeric antigen receptor, targeting for intracellular drug delivery etc. And figure 2 is a very well presented schematic diagram of ERBB2 pathway. However there are some minor comments the authors could address and include in this review.

One of the well-studied targeted therapeutic for ERBB2/Her2 is a small peptide molecule KCCYSL. There are many literatures available which have used this to target drugs to Her2 receptor (https://doi.org/10.1016/j.msec.2016.05.014). The authors could include this in this review.

Another aspect which is currently missing and would make the review more interesting is to add the aspect of using various targeted nanomedicines for CRC receptors .
